# Orangutans’ Comprehension of Zoo Keepers’ Communicative Signals

**DOI:** 10.3390/ani9060300

**Published:** 2019-05-31

**Authors:** Guillaume Dezecache, Aude Bourgeois, Christophe Bazin, Philippe Schlenker, Emmanuel Chemla, Audrey Maille

**Affiliations:** 1Institut Jean Nicod, Département d’Etudes Cognitives, ENS, EHESS, PSL Research University, CNRS, 29 rue d’Ulm, 75005 Paris, France; guillaume.dezecache@gmail.com (G.D.); philippe.schlenker@gmail.com (P.S.); 2Department of Experimental Psychology, University College London, London WC1E 6BT, UK; 3Ménagerie du Jardin des plantes, DGD Musées, Jardins et Zoos, Muséum National d’Histoire Naturelle, 57 rue Cuvier, 75005 Paris, France; aude.bourgeois@mnhn.fr (A.B.); christophe.bazin@mnhn.fr (C.B.); 4Department of Linguistics, New York University, 10 Washington Place, New York, NY 10003, USA; 5Laboratoire de Sciences Cognitives et Psycholinguistique, Département d’Etudes Cognitives, ENS, PSL Research University, EHESS, CNRS, 29 rue d’Ulm, 75005 Paris, France; chemla@ens.fr; 6Unité Eco-anthropologie UMR 7206, Muséum National d’Histoire Naturelle, CNRS, Université Paris Diderot, 17 place du Trocadéro, 75116 Paris, France

**Keywords:** animal welfare, medical training, gestures, speech, inter-species communication, orangutans

## Abstract

**Simple Summary:**

Most modern zoos work towards the promotion of captive animals’ welfare. One way of achieving this is by encouraging cooperative interactions between keepers and zoo animals, for instance during sessions of conditioning training. To be most effective, communication between them should use those channels that are most useful to the animals. In this study, we asked whether captive orangutans were capable of understanding keepers’ instructions when they were employing words only, gazes only, gestures only, or all signal types combined. Our results indicate that the subjects only need gestures to respond to the keepers’ instructions. In two other experiments, we examined why gestures were so effective. One hypothesis was that they resemble what they refer to. However, we found no indication that gestures providing iconicity or even pointing could help orangutans acquire associations between a new gesture and referent. Our results suggest that, among human communicative signals, captive orangutans would prefer gestures. Why this is the case should be the focus of future research.

**Abstract:**

Zoological institutions often encourage cooperative interactions between keepers and animals so as to promote animals’ welfare. One useful technique has been conditioning training, whereby animals learn to respond to keepers’ requests, which facilitates a number of, otherwise sensitive, daily routines. As various media have been used to convey keepers’ instructions, the question remains of which modality is best to promote mutual understanding. Here, we explored this question with two captive female orangutans. In the first experiment, we compared orangutans’ understanding of previously acquired instructions when those were performed with verbal signals only, gazes only, gestures only, and when all those modalities were combined. Our results showed that gestures were sufficient for successful comprehension by these two apes. In the second experiment, we asked whether this preference could be driven by the non-arbitrary relationship that gestures bear to what they refer to, through iconicity or pointing. Our results revealed that neither iconicity nor pointing helped the subjects comprehend the keepers’ instructions. Our results indicate a preference for instructions given through gestural signals in two captive female orangutans, although its cause remains elusive. Future practice may encourage the use of gestures in communication between keepers and orangutans in general or potentially other animals.

## 1. Introduction

Zoological institutions housing hominoids must ensure that management practices meet the highest level of welfare. To meet this aim, one strategy has been to encourage safe cooperative interactions between keepers and animals. In this context, conditioning training with positive reinforcement has become widespread [1,2,3,4,5]. In conditioning training with positive reinforcement, animals progressively learn how to solve problems, such as responding to requests from zoo keepers [6]. When a request is produced, and the animal displays the desired behavior, a secondary reinforcer (e.g., a click) and a primary reinforcer (e.g., food) are used to reward the animal. One example of reinforced behavior is the presentation of a particular body part by the animal after the keeper produces a signal. This technique, based on voluntary participation from the animals, has several advantages; for example, it facilitates husbandry procedures (e.g., shifting animals between enclosures or requesting an object that has been caught by the animal) [7,8,9]. It has also made veterinary procedures (e.g., clinical examination, wound care, auscultation, ultrasound, intramuscular injections) safer both for the keeper and the animal since training allows them to be carried out without anesthesia or restraint [2,10,11,12,13,14,15]. This technique is sometimes considered as part of the cognitive enrichment toolkit [16].

Successful training programs may use requests that combine a variety of cues by keepers [17]. Which signals should keepers use to optimally communicate their requests to captive animals? Which is the modality or modalities that matter most to the animal? Responses to these questions are species-dependent but together they are crucial to the management of captive animal populations.

A range of studies have evaluated animals’ ability to learn and understand human signals, particularly in great apes. Investigations on whether apes are capable of learning to produce human sounds and use them in dialogue-like situations have resulted in failure [18]. Instead, researchers tried to teach gestures (inspired by sign language) to ape species [18,19,20,21]. While considerable doubt has been cast on the productivity of the rule system that was learnt [22,23], it appears that apes exhibited fluency with the acquisition and/or comprehension of signals employing gestural modality [19], a conclusion consistent with a growing body of research on the rich inventory of gestures naturally produced by apes [24,25,26,27,28,29,30,31]. Still, apes also display rich, if ill-understood, vocalizations. There is also evidence in some primate species of an ability to understand the acoustic signals of other species (e.g., [32,33,34]). Was the advantage of gestures over speech sounds in language acquisition entirely due to articulatory constraints, or is there a more general preference for gestural communication that could be evidenced in perception as well? If a preference is observed for gestural modality, which role does iconicity, i.e., the non-arbitrary link between the form of a signal and the object that it refers to, play in this acquisition process? In humans (and by contrast to chimpanzees), iconicity can play an important role in successfully referring to objects [35,36]. A preference for gestural modality has been found in other species, in particular, domestic species such as dogs trained to respond to humans’ verbal and gestural signals [37]. If indeed the animal species under investigation prefers gestural modality, future training should focus on this particular signaling media.

This study focused on Bornean orangutans (*Pongo pygmaeus*). At the ‘Menagerie, le zoo du Jardin des Plantes’ (Museum National d’Histoire Naturelle, Paris, France), orangutans have been involved in medical training since 2005. Currently, four orangutans (one male and three females) are trained to respond to requests from keepers.

In this study, we aimed at understanding which types of signals are more easily dealt with by the orangutans of our study site. Specifically (Experiment 1), we looked at which modality was most efficient in conveying the requests produced by the keepers, by studying the relative understanding of signals produced using one out of the three potential modalities typically employed during training (voice, gesture, and gaze). Additionally (Experiment 2), we sought to evaluate the role of signal arbitrariness (whether a signal bears resemblance with its referent or not) (Experiment 2—Phase A) and pointing (Experiment 2—Phase B) in orangutans’ comprehension of keepers’ requests.

## 2. Material and Methods

### 2.1. Ethics

This study was approved by the Comité d’Ethique Cuvier N° 68 (DAECC 68-102). Orangutan subjects were separated from the dominant individual of the group during testing (as in during the daily training routine), but we interrupted the procedure if orangutans showed signs of distress.

### 2.2. Animal Subjects

Two captive-born Bornean orangutans (*Pongo pygmaeus*), 29-year-old adult female Theodora (born in 1988, 2170-EAZA) and her 13-year-old daughter Tamü (born in 2004, 3164-EAZA) were observed and tested (i.e., ages when the study started). They were housed with another adult female and an adult male in five indoor enclosures (2.5 m × 5.5 m × 4.5 m) with controlled access to an outdoor area (13 m × 6.5 m × 4.5 m) at La Ménagerie, le zoo du Jardin des Plantes at the Museum National d’Histoire Naturelle (Paris, France). Orangutans are fed every day at 9:00 a.m., 10:30 a.m., 12:00 a.m., 2:30 p.m. and 5:00 p.m with Mazuri^®^ (Dietex France, Argenteuil, France) primate maintenance biscuits, fruits, vegetables, and dairy products. They receive enrichments (feeding, physical, and cognitive) on a daily basis.

It is important to note that the two other orangutans of La Ménagerie could not be recruited for participation over the course of this study: One individual was relatively old and participation could cause her tiredness; the other individual was relatively new to the site and was not proficient enough at the medical training.

### 2.3. Medical Training

Theodora and Tamü have been involved in a medical training program since their transfer at la Ménagerie in 2007. They participate in training sessions on a daily basis. The medical training consists of a series of 29 requests (Appendix A). In part of those requests, animals are expected to present the body part that is being requested by the keeper, by making contact between the requested body-part and the tip of a spoon held by the keeper (the target). When they are successful at presenting the requested body part, the keeper produces a clicker sound as a secondary reinforcer and gives a reward (e.g., a piece of fruit) and verbal praise. If they do not succeed, no reward is given, and the keeper may repeat the same request or ask for another. Requests are multimodal signals. They are expressed using a combination of French words (corresponding to the body area or task that is being requested) and other verbal and vocal cues (encouragements), gazes (the keeper may look at the body area or object that is being requested), and gestures (the body posture and the movement and location of the hands and target—the spoon—and their final position may help the animal to understand which body part is being requested for).

### 2.4. General Note on the Procedures

Theodora and Tamü were tested at times when they usually participate in medical training sessions (between 4:00 and 5:00 p.m. in winter, and between 5:00 and 6:00 p.m. during summer). Only one zoo keeper (C.B.) was involved in the entire study, to reduce potential variation in the quality and quantity of the signals used for requests within and across experiments.

### 2.5. General Note on the Statistical Analyses

All statistics were performed with R v.3.5.0 [38]. The significance level was set at 0.05. Mixed generalized linear models were constructed using the glmer function in the *lme4* package [39], and statistical tests were performed using the Anova function in the *car* package which calculates analysis-of-variance tables based on Type-II Wald tests, even for fixed effects in generalized linear mixed-effects models [40]. Post-hocs tests with a Tukey correction were performed using the emmeans function in the *emmeans* package [41]. Considering that only two subjects were involved in the study, we decided to fit some models with Subject as the fixed effect rather than random factor and to aggregate the data across sequences and sessions to avoid any risk of pseudo-replication. Some models did not converge or achieved boundary fits only, which, in some occasions, led us to simplify them as reasonably as possible. All data and scripts for analyses with details and justifications are available at: https://osf.io/hjcfz/. We note that running the analysis on each subject separately yielded the same qualitative results.

## 3. Experiment 1

In the first experiment, we sought to identify the respective contribution of each modality combined with the signals produced by the human zoo keepers in supporting the subjects’ understanding of requests, as they normally occur in the medical training set up at the study site.

### 3.1. Procedure

Experiment 1 was conducted between 12 and 28 November 2017. Each of the two orangutan subjects was tested in six experimentally-modified training sessions, on six different days. Within a session, subjects were exposed to 15 requests repeated across six sequences. Those 15 requests were selected from the set of requests subjects are typically exposed to during normal training sessions (Appendix A). Selection ensured that all requests were well known to the subjects (i.e., all 15 requests were considered to be ‘learnt’ by the subjects, as judged by the medical training team and based on the responsiveness of the animal in training sessions). It was also ensured that all requests unambiguously referred to one single body area the subject had to present to the keeper.

During a session, each orangutan was exposed to a total of 90 requests (6 sequences × 15 requests). The 15 requests were expressed in the same order for each sequence within a session, but the order of the requests was pseudo-randomized between sessions.

A session always started with a sequence in the Control condition, which resembled the normal medical training: The keeper expressed each request to the orangutan using the usual combination of modalities (verbal and gestural signals and gazes) and rewarded each correct behavioral response with a clicker sound (secondary reinforcer) followed by a piece of apple and verbal praise (Table 1). The Control condition was performed twice per session. The aim of implementing those two sequences of requests in the Control condition was to sustain the motivation of the subjects and make sure our experimental modifications did not alter their motivation to participate in medical training in the future. Only responses to the four other sequences of each session were analyzed, which are presented below.

A sequence in the All condition also resembled the normal medical training (as all modalities were used in combination), except that the correct behavioral responses were rewarded with the clicker sound only (Table 1). Note that no food reward was provided. This was done to avoid over-feeding the animals during the experiment. Indeed, 100% success would have required us to reward the orangutan subjects 90 times (an amount of food which is beyond the needs of the animal). In addition, whereas in the normal training regime the keeper may repeat some of the signals if needed, only one signal was given in each of the three modalities (one gesture, one word, one gaze only).

In the three remaining sequences, only one modality was used by the keeper to express requests to the orangutan (Table 1). In the sequence for the Words condition, the keeper uttered the usual signal (French word) referring to the body part that the orangutan was requested to present (e.g., ‘ton ventre’, which is French for ‘your belly’, to request the orangutan to present the belly). In the sequence for the Gestures condition, the keeper used gestures referring to the body part that the orangutan was requested to present (e.g., pointing the target towards the belly to request presentation of the belly). In the sequence for the Gazes condition, the keeper looked towards the body part that the orangutan was supposed to present. Note that each signal was only used once per trial (i.e., one word was uttered in the Words condition; one gesture was produced in the Gestures condition; one gaze movement in the Gazes condition). Correct responses in these three testing conditions were rewarded with the clicker sound only. For each of these three conditions, cues from the irrelevant modalities were blocked using sunglasses (to eliminate gaze cues in conditions Words and Gestures), a dummy verbal request (the word ‘voiture’, which is French for ‘car’ and is never used with the orangutans, to eliminate verbal and vocal cues in conditions Gestures and Gazes), or by ensuring that the same movement of the target was used across requests (to eliminate gestural cues in conditions Words and Gazes). Video extracts showing the procedures for condition Words and Gestures are available in Appendix A.

Orangutans were allowed a maximum of 5 s to produce the correct behavioral response. If they failed to respond successfully within 5 s, the following request was expressed. At the end of each sequence, subjects received a piece of apple, to help keep their motivation sustained. Sessions were videotaped by a research assistant (G.D.) whose role was also to inform the keeper (C.B.) when the 5 s delay allowed to orangutans for responding had elapsed. Note that the order of requests and conditions was provided to the keeper on a sheet of paper taped on the wall at eyes level.

### 3.2. Data Coding

Success was coded by the keeper (C.B.). One of the authors (G.D.) re-coded independently 16.5% of the trials by watching the videos and the agreement was found to be almost perfect (*k* = 0.98). The initial coding of the keeper was used for the analyses.

### 3.3. Results

Tamü was overall successful in 92.1 ± 11.1% (mean ± standard error) of the trials in the All condition, 84.4 ± 14.9% of the trials in the Gestures condition, 8.9 ± 11.7% in the Words condition, and 6.7 ± 10.2% in the Gazes condition. Theodora was successful in 82.2 ± 15.7% of the trials in the All condition, 75.6 ± 17.6% of the trials in the Gestures condition, 11.1 ± 12.9% in the Words condition, and 7.8 ± 11.0% in the Gazes condition.

Results are represented per session (Figure 1) and per request (Figure 2). In all cases, an advantage of the All and Gestures conditions over the Words and Gazes conditions is apparent.

We fit a mixed generalized linear model for binomial data with Success as the dependent variable, with Condition, Subject, and their interaction as fixed effects, and with a random intercept for Request (body parts) and Item number (to distinguish between trials from different sequences and sessions); all details and justifications are provided in the Appendix A at https://osf.io/hjcfz/. The results showed a significant main effect of Condition ($\chi^{2}$(3) = 226, *p* < 0.001), no significant main effect of Subject ($\chi^{2}$(1) = 2.52, *p* = 0.11), and no significant interaction effect ($\chi^{2}$(3) = 4.61, *p* = 0.20). In a post-hoc comparison of the conditions, we observed no significant difference between the All and the Gestures conditions (z = 2.05, *p* = 0.17) or between the Gazes and Words conditions (z = −0.95, *p* = 0.78), but there were significant differences between either of the former two conditions (All and Gestures) and any of the latter two conditions (Gazes and Words): All zs > 11.3, *ps* < 0.0001. (This model only achieved a ‘boundary fit’; however, the effects were confirmed using a simpler model based on data aggregated across sequences and sessions, simply using Condition as a main effect, and traditional random intercepts for Subject and Request).

### 3.4. Discussion

Our results indicate that when the keeper only used gestures, the two orangutan subjects were capable of presenting the correct body area around 80% of the time. This is considerably higher than their performance when responding to keepers’ requests when only words and gazes were used (around 10% and 7%, respectively). This result appears to have remained stable across testing days (Figure 1) and suggests that this preference was not built during the course of the experiment but existed prior to our intervention. This preference for gestures over words and gazes can be observed in both subjects. Although it is tempting to conclude that all orangutans share this preference, it is to be remembered that Theodora and Tamü are kin. Their preference may therefore have a genetic or ontogenetic basis. Besides, this preference for gestures might be specific to orangutans of la Ménagerie, le zoo du Jardin des Plantes, where the two studied orangutans were subjected to medical training that is likely specific to the study site.

Exceedingly poor performance when either words or gazes were used in isolation may appear surprising. As training is always performed with the same verbal information across requests, Theodora and Tamü should have been able to learn associations between a given word and the presentation of a body part from taking part in the medical training. The fact that words specifically appear to have not been learnt suggests that they ignored this information in the past and focused on gestural signals. The performance of Theodora and Tamü was also strongly impaired when only gaze cues were provided. It is known than orangutans can gaze-follow [42,43,44], but our results may be linked to a lack of attention to the eyes of the keeper during the experiment, an explanation which is plausible in view of reports on orangutans’ avoidance of mutual gaze [45].

A potential reason for the higher performance for requests of the Gestures condition compared to the Words condition is that the signal used in the Words condition changes each time (a different word for each body part). On the other hand, the gestural signal remains exactly the same, with only the location of the gesture changing across requests. It should be noted, however, that gestures not directly employing indication of a location (i.e., ‘tongue’ and ‘teeth’, see Appendix A for details on those requests) were also much more successfully addressed in the Gestures condition (Figure 2). Second, the Gazes condition is similar in this respect to the Gestures condition (the only change across requests is the target of the gaze). However, the Gazes condition was associated with poor performance overall.

## 4. Experiment 2

Signals are multimodal in ecological situations, as this has been increasingly acknowledged [46,47,48,49]. Why did subjects mostly rely on gestural signals to respond adequately to requests from their zoo keeper? A first possibility is that gestures used in training, contrary to words, are ‘iconic’; that is, their forms have a non-arbitrary connection with what they refer to (the body area that is being requested for presentation). For instance, the gestural signal used as the request for the orangutan to open the mouth and present the tongue consists of sticking one’s tongue out. The link between iconic (rather than arbitrary) signals and a referent can facilitate acquisition, at least in humans [35,36].

A second possibility is that the relative ease of gesture comprehension was driven by some of the gestures used, namely those that involve ‘pointing’, i.e., the body part which is being requested for presentation is physically indicated by the position of the hand and finger of the keeper, and physical distance is reduced between the hand/finger of the keeper and the body part being requested. Comprehension of pointing can be acquired in apes [50,51,52].

Both possibilities were examined here, in separate phases (Phases A and B). In Phase A, we asked whether subjects could learn an *arbitrary* association between a gesture and a referent (more easily than one between a word and a referent). In Phase B, we asked whether subjects could better learn from an iconic association between a gesture and a referent, or from a pointing gesture.

### 4.1. Procedure

Phase A of Experiment 2 was conducted between 4 May and 11 June 2018. Phase B of Experiment 2 was conducted between 7 September and 8 October 2018. For Phase A, subjects were Tamü and Theodora. For Phase B, only Tamü could participate, as Theodora was found to be pregnant and the veterinary team judged that participation could be stressful and cause tiredness.

In both phases of Experiment 2, subjects had to choose one stimulus among the two stimuli that were being shown simultaneously to them, with each stimulus being associated with a specific signal produced by the keeper; signals were unknown to the subjects. Subjects made a choice by producing a pointing gesture towards the stimulus of their choice.

The stimuli were cardboards (of dimensions 15 × 6 cm) that were placed on each extremity of a horizontal 70 cm long wooden perch, the distance between the two stimuli being 50 cm. The wooden perch was positioned on a 150 cm high tripod so that the stimuli could be presented at an orangutan-chest level (Figure 3).

In Phase A, subjects were exposed to an Arbitrary–Word condition and to an Arbitrary–Gesture condition (Figure 4). In both conditions, the association between signal and referent was totally arbitrary. In the Arbitrary–Word condition, the keeper produced either a ‘bebebe’ or a ‘dadada’ verbal signal, and subjects had to learn that ‘bebebe’ was associated with a round-shaped stimulus whereas ‘dadada’ was associated with a triangle-shaped stimulus (Figure 4). In the Arbitrary–Gesture condition, the keeper produced either a ‘high five’ (open right hand with the palm towards the subject) or a ‘thumbs up’ (closed right hand except the thumb finger that is erected) gestural signal, and subjects had to learn that ‘high five’ was associated with a cross-shaped stimulus whereas ‘thumb up’ was associated with a square-shaped stimulus (Figure 4). To make the experiment simpler for the keeper (who had to reward correct answers with a click), we did not counterbalance the association between signal and stimulus between subjects.

In Phase B, the subject was exposed to an Iconic–Gesture condition and to a Pointing–Gesture condition (Figure 4). In both conditions, the association between signal and referent was relatively non-arbitrary. In the Iconic–Gesture condition, the keeper produced either a letter A from American Sign Language fingerspelling (hereafter, ASL) or letter V from ASL. The subject had to learn that letter A from ASL was associated with a stimulus depicting the shape and drawing of a right-hand producing the letter A from ASL, whereas letter V from ASL was associated with a stimulus depicting the shape and drawing of a right-hand producing the letter V from ASL (Figure 4). In the Pointing–Gesture condition, the keeper produced a pointing gestural signal toward the stimulus located either on the left or right-side of the perch (Figure 4). The subject had to learn that the direction of the pointing indicated the correct response, no matter which stimulus it was (either a stimulus depicting the shape and drawing of a right-hand producing the letter B from ASL or a stimulus depicting the shape and drawing of a right-hand producing the letter L from ASL, Figure 4).

In both phases, trials consisted of the keeper producing the signal. Once the keeper thought that the subject had heard or watched the signal, the tripod was tilted towards the subject for her to be able to make a choice by pointing towards one of the cardboard stimuli. To control for the influence of gaze cues, the keeper wore sunglasses during all the sessions. Moreover, in the conditions involving Gestures, the keeper always produced the gestured-stimulus with the right hand, holding the hand at an equal distance between the two stimuli to control for the potential influence of positional cues. Subjects were allowed a maximum of 5 s per trial to point towards the correct cardboard stimulus. If they failed to respond successfully within 5 s, either because they pointed toward the stimulus that was not associated with the signal or because they did not provide any response, the following trial was performed. When subjects responded successfully within the 5 s, they were rewarded with verbal praise, a click and a piece of apple (with a maximum of 20 pieces of apple per session).

Each phase consisted of 10 sessions. During a session, the subject was exposed to two different conditions. Each condition was used for sequences of 10 consecutive trials, twice per session. Each sequence contained five requests of each of the two kinds in the relevant condition, distributed in random order over the sequence. In total, there were 40 trials per session (2 sequences × 10 trials × 2 conditions). The order of the sequences was randomized between sessions. The position of the associated stimulus was fixed within a sequence but counterbalanced among two sequences in the same session and condition.

### 4.2. Pilot: shaping of Orangutans’ Pointing

Shaping was used to teach Tamü and Theodora that pointing to the cardboard was the type of response that was desirable. Both subjects were presented with 10 sessions of six trials, with a rectangle-shaped blank cardboard attached either to the left or right side (pseudo-randomized order) a same number of times per session. The wooden perch was tilted towards the subject for 5 s. Subjects were considered to have learnt that pointing to the cardboard was the expected response if they were successful in 10 consecutive trials (pointing to the cardboard at least one time within the 5 s). After each correct response, subjects were rewarded with a click and a piece of apple. It took 11 trials for Tamü and 49 for Theodora to understand the task.

### 4.3. Data Coding

Since agreement between the keeper (C.B.) and the research assistant (G.D.) was almost perfect in Experiment 1, and Experiment 2 could not be videotaped because the corridor used for running the experiment could only accommodate one research assistant and one keeper. Since agreement between the keeper (C.B.) and the research assistant (G.D.) was almost perfect in Experiment 1, we judged the live coding of the keeper to be sufficiently reliable, and relied on it for both phases of Experiment 2. The assistant (G.D. in Phase A and A.M. in Phase B) were involved in both monitoring time and noting down whether the subject was correct, following the decision of the keeper.

### 4.4. Results

In analyzing the data of Experiment 2, we considered a trial as successful only when the first response provided by the subject was correct (pointing directed toward the stimulus associated with the signal). An orangutan was unsuccessful in a trial if her first response was incorrect or because she did not respond at all.

#### 4.4.1. Phase A: Arbitrary Words and Arbitrary Gestures

Tamü was successful in 46.5 ± 15.8% (mean ± standard error) of the trials in the Arbitrary–Gesture condition and 41.5 ± 15.6% of the trials in the Arbitrary–Word condition. Theodora was successful in 41.5 ± 15.6% of the trials in the Arbitrary–Gesture condition and 43.0 ± 15.7% of the trials in the Arbitrary–Word condition. Results were thus overall very poor. They were hard to distinguish from chance levels, and there was little difference between conditions. Note that the chance level was below 50% since participants could provide a correct response, an incorrect response, but also no response at all if they so decided. Tamü did not respond at all in 40.5% of the trials in the Arbitrary–Gesture condition and 36.0% of the trials in the Arbitrary–Word condition. Theodora did not respond in 52.5% of the trials in the Arbitrary–Gesture condition and 55.0 % of the trials in the Arbitrary–Word condition. Figure 5 represents the results per session for each individual.

We fit a mixed generalized linear model on aggregated data across sequences and items with average Success as the dependent variable, with Condition, Subject, and their interaction as fixed effects, and with a random intercept for Session (all details are provided in the Appendix A at: https://osf.io/hjcfz/). The results showed no significant main effect of Condition ($\chi^{2}$(1) = 0.19, *p* = 0.66), no significant main effect of Subject ($\chi^{2}$(1) = 0.19, *p* = 0.66), and no significant interaction ($\chi^{2}$(1) = 0.67, *p* = 0.41).

Importantly, both subjects showed a significant preference for the square stimulus above the cross stimulus in the Arbitrary–Gesture condition (Binomial test: Tamü: *p* < 0.001; Theodora: *p* = 0.023). This second preference could be linked to the pilot phase, where orangutans were presented with a rectangle-shaped stimulus. In contrast, there was no difference in the number of choices of the round and triangle stimuli in the Arbitrary–Word condition (Binomial test: Tamü: *p* = 0.113; Theodora: *p* = 0.142).

#### 4.4.2. Phase B: Iconic Gestures and Pointing Gestures

Tamü was successful in 50.5 ± 15.9% (mean ± standard error) of the trials in the Iconic–Gesture condition and 51.5 ± 15.8% of the trials in the Pointing–Gesture condition. In this phase, Tamü did not respond in only 0.5% of the trials both in the Iconic–Gesture and the Pointing–Gesture conditions. Figure 6 represents her results per session. We fit a mixed generalized linear model on aggregated data across sequences and items with average Success as the dependent variable, with Condition as a fixed effect, and with a random intercept for Session (all details are provided in the Appendix A at: https://osf.io/hjcfz/). Figure 6 represents the results per session for this individual.

The results showed no significant main effect of Condition ($\chi^{2}$(1) = 0.04, *p* = 0.84). We noted that when there was a noticeable difference, there seemed to be some advantage of the pointing condition over the iconic condition appearing in Sessions 2 and 3. When we restricted the analyses to these sessions, the chi-square test showed that this did not reach significance after correction for multiple comparisons (*χ*^2^(1) = 4.5, *p* = 0.035, before correction).

Interestingly, Tamü showed a preference for the letter L (from ASL) stimulus above the letter B (from ASL) stimulus in the Pointing–Gesture condition (Binomial test: *p* = 0.002), whereas there was no difference in the number of choices of letter A and V (from ASL) in the Iconic–Gesture condition (Binomial test: *p* = 0.887).

### 4.5. Discussion

Experiment 2 was carried out to understand why subjects exhibited a higher performance when exposed to gestural signals in Experiment 1.

In Phase A, we evaluated orangutans’ performance as they had to learn arbitrary associations between a visual stimulus and either a gesture or a vocalization. Results showed no evidence of learning of those associations by either subject in either condition. There was some indication that Tamü was more successful at the Arbitrary–Gesture vs. Arbitrary–Word condition, but this was confined to Testing Day 4. This pattern could indicate some momentary understanding of the task, but it is yet difficult to understand why she would have then lost this comprehension. A more conservative conclusion of Phase A is that 10 sessions are not sufficient for otherwise trained orangutans to pick up an arbitrary association between a stimulus and a gesture or a verbal sound.

Phase B was designed to look at learning of non-arbitrary associations between a gesture and a visual stimulus. Results from Phase B showed no learning of associations. An inspection of the performance across testing days showed that Tamü performed well in the Pointing–Gesture condition at Testing Days 2 and 3 (circa 80% of correct responses) but her performance dropped later on. In comparison, no such pattern was visible in her performance in the Iconic–Gesture condition. In those two early sessions, it seems that Tamü was able to represent the association between a pointing gesture and the object being pointed at. Yet, statistical analysis failed to show an interaction between the condition and the number of testing days, which prevents us from drawing strong conclusions from the increase in performance early on in the Pointing–Gesture condition.

## 5. General Discussion

Recent zoo management practices have emphasized the importance of positive interactions between zookeepers and animals. In this context, conditioning training with positive reinforcement has appeared to facilitate a great number of procedures [7,8,9]. The efficiency of conditioning training is dependent on the use of appropriate signals by the keepers, i.e., signals that can be easily understood by the animals. Determining which modality of communication is preferred by a given species (or even individual) is thus critical to improving medical training techniques and fostering animals’ well-being.

In our study, we took advantage of the well-designed medical training technique developed since 2005 on Bornean orangutans (*Pongo pygmaeus*) at the ‘Ménagerie, le zoo du Jardin des Plantes’, in Paris. Specifically, seven orangutans have been involved in the medical training program in this zoological institution; two of them could be tested. We first asked whether they were able to understand keeper requests from the normal medical training sessions when requests were produced with or without gestures, words, and/or gazes. We found that performance was very high when all modalities were used. When gestures only were used, performance remained high whereas when words or gazes were used in isolation, performance was poor. Our results are in line with a preference for the gestural modality (vs. the vocal modality) in studies examining the acquisition of signals used by humans in apes [18,19,20,21]. Our findings are also consistent with similar reports on dogs trained with verbal and gestural cues [37,53,54]. The fact that gaze cues did not lead to correct responses by the subjects may be surprising given orangutans’ ability to gaze-follow [44], but this might be due to orangutans’ tendency to avoid mutual gaze [45]. Our results suggest that gestural modality seems to be appropriate and efficient for comprehension of the zoo keepers’ requests, at least with the study species and individuals of the study site. Such a preference for using gestural cues performed by humans during training sessions appears consistent with the theory of a gestural origin of language, which assumes that language arose from gestural rather than vocal communication [55,56,57]. One should note, however, that our study was based on two individuals trained at a specific site and with one specific keeper. Although we used a modelling approach suited to analyzing responses of two individual subjects, statistical analyses would be more powerful with additional subjects. Ideally, our study should be replicated in other settings and in other sites, possibly with a larger cohort.

Why should the gestural modality be preferred over verbal signals and gaze signals in those orangutans? It is possible that orangutans pick up on gestural signals they are presented with because of their resemblance with signals that are part of their natural repertoire. But if one expects animals to prefer signals that are also used during intra-specific communication events, gazes should have been preferred, as the referent of the gaze is the object being looked at and attended to. This is not what we found. Another possibility is that orangutans prefer gestural signals because those do not bear an entirely arbitrary relationship with what they refer to, with gestures used by keepers either presenting a certain degree of iconicity (the signal resembles the referent) or being deictic (the signal points towards the referent, which is then the object that is close to the tip of the keepers’ finger). The association between a gesture and a referent should thus be considerably easier to draw and learn.

Experiment 2 examined this hypothesis. We looked at whether signals associated with referents in an arbitrary or non-arbitrary fashion could be learnt by the subjects. In the first phase of the experiment, we found that when the associations between signal and referent were arbitrary, learning did not occur for the subjects, at least in the first 10 sessions. In the second phase of the experiment, we found that when the associations between signal and referent were non-arbitrary (here: Gestures that are iconic or deictic with their referent), learning did not clearly happen either in the 10 available sessions, although there may have been an indication that pointing could have provided an advantage for one of our subjects. The absence of learning for the relationship between iconic signal and their reference appears consistent with previous studies on apes examining the comprehension of iconic signals [35,36]. However, it should be recalled that iconicity was here defined as a similarity between signal and referent from the perspective of human experimenters. For orangutans, two-dimensional representations of human hands may not be associated with actual human hands. In fact, all iconicity is not created equal, as some modalities may naturally bear more iconicity than others. Iconic gestures are indeed better understood than iconic vocalizations in human children [34].

Our study is therefore inconclusive in clarifying why gestures are preferred by orangutans, simply because of a lack of learning through the limited experimental sessions of Experiment 2. Failure to master the associations between gestural signals and their referents in Experiment 2 (by contrast with Experiment 1) may have several sources. First, in Experiment 1, the gestures had already been learnt by the animals. Given the cognitive abilities of orangutans [58,59,60,61,62], it is very likely that the associations provided in Experiment 2 would have been learnt eventually. Second, the correct responses to the gestural requests used in Experiment 2 involved touching an external object. Since the bulk of interactions between keepers and orangutans are associated with the presentation of one’s body part by the animal, the subjects may have had difficulty understanding that a request could refer to an external object. It should nonetheless be kept in mind that some of the requests used by keepers required the animal to fetch an object (see Appendix A). Furthermore, we used ‘shaping’ before Experiment 2, to foster the sort of responses we expected in Experiment 2 (i.e., touching an external object rather than presenting a body part).

More generally, Experiment 2 may have not been appropriate to induce learning in orangutans at all, regardless of the type of requests used. First, animals were allowed 5 s to produce the correct response (a procedure that mimics the classical medical training used at the study site). Rewards given to correct responses could therefore ambiguously be related to any of the responses produced by the orangutans, including incorrect ones. In addition, the task was novel and learning was limited (10 sessions of 20 trials only per condition). Future investigations willing to use the procedure developed in Experiment 2 should include more subjects, more frequent and longer sessions overall, and more consistent rewarding (e.g., reward only if the first response is the correct one).

Despite the clear limitations of Experiment 2, our study points to gestural signals as a potential key candidate to set up a communication system during cooperative interactions between zoo keepers and captive orangutans. Drawing a stronger conclusion would require replication of our experiment with orangutans from other sites. Ideally, our approach could also be applied to other species. The results generated could ease the implementation of positive reinforcement learning procedures with a variety of species and foster the relationship between zoo keepers and animals in their care.

## 6. Conclusions

Our results point towards a preference for human gestural signals by orangutans, a conclusion which has potential to improve communication between keepers and captive orangutans during cooperative interactions such as medical training, provided this is also evidenced in other sites. Ideally, our results should be replicated in other sites and with a larger sample. The question of what it is exactly in gestures that helps mutual comprehension between keepers and orangutans remains elusive and should be the focus of future research.

## Figures and Tables

**Figure 1 animals-09-00300-f001:**
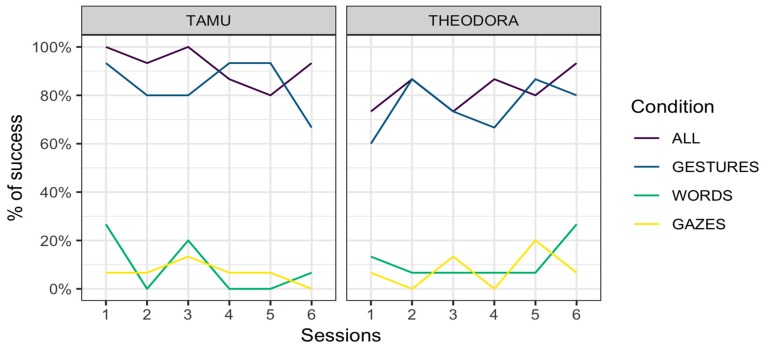
Performance for the two subjects across conditions and sessions.

**Figure 2 animals-09-00300-f002:**
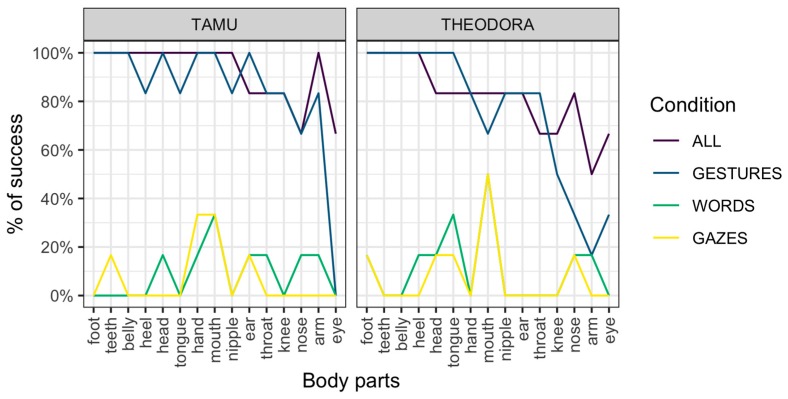
Performance for the two subjects across conditions and requested body parts.

**Figure 3 animals-09-00300-f003:**
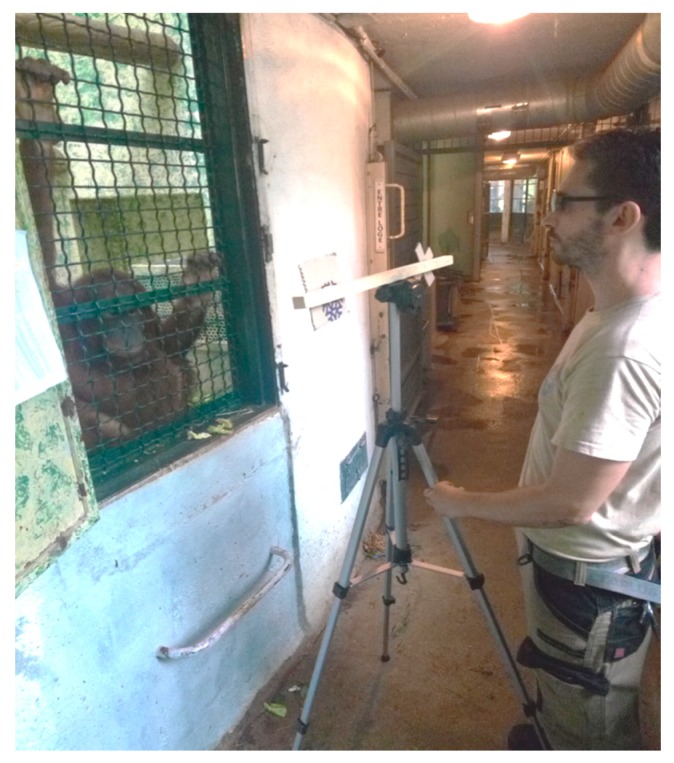
Apparatus used for Experiment 2. The picture shows the wooden perch with the two shaped cardboard stimuli being presented on the extremities. The zoo keeper presents the subject with the apparatus and stimuli.

**Figure 4 animals-09-00300-f004:**
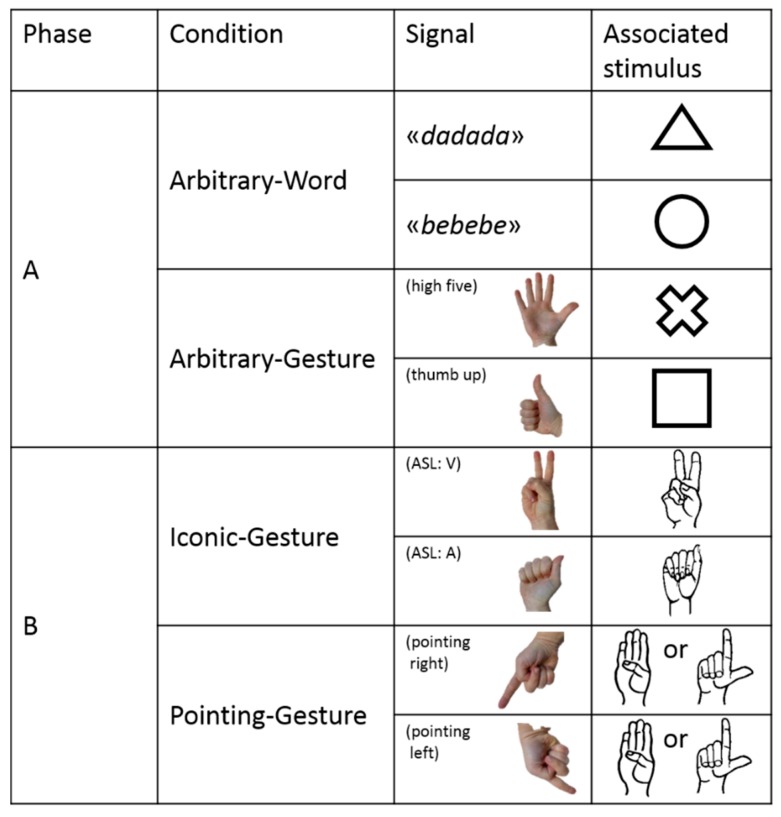
Signals and associated stimuli used for each phase and condition of Experiment 2.

**Figure 5 animals-09-00300-f005:**
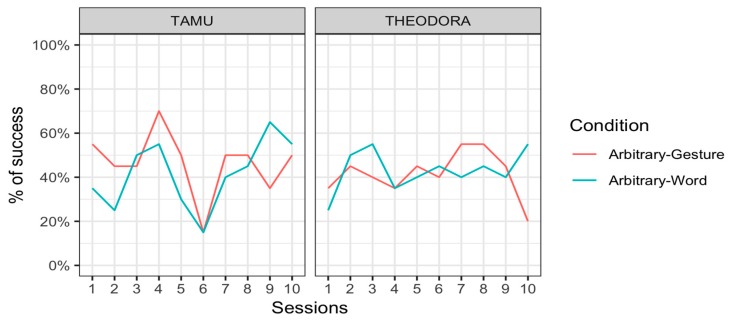
Percentage of successful responses (mean) of the orangutans Tamü (left) and Theodora (right) in the Arbitrary–Gesture condition (pink line) and the Arbitrary–Word condition (blue line) for the 10 sessions.

**Figure 6 animals-09-00300-f006:**
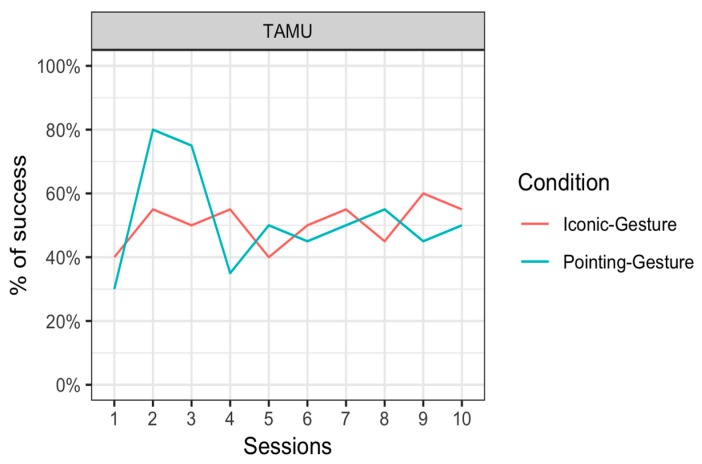
Percentage of successful responses (mean) of the orangutan Tamü in the Iconic–Gesture condition (pink line) and the Pointing–Gesture condition (blue line) in the 10 sessions.

**Table 1 animals-09-00300-t001:** Modality of the request and reward types used in the Control, All, Words, Gestures, and Gazes conditions of Experiment 1.

Condition	Modalities Composing the Request			Reward (if Correct Response)		
	Verbal	Gestural	Gazes	Clicker	Piece of Apple	Verbal Greetings
CONTROL	Yes	Yes	Yes	Yes	Yes	Yes
ALL	Yes	Yes	Yes	Yes	No	No
WORDS	Yes	No (static spoon)	No (sunglasses worn)	Yes	No	No
GESTURES	Dummy word (“voiture”)	Yes	No (sunglasses worn)	Yes	No	No
GAZES	Dummy word (“voiture”)	No (static spoon)	Yes	Yes	No	No

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
