# Peer review of "Orangutans’ Comprehension of Zoo Keepers’ Communicative Signals"

_animals, 2019, doi:10.3390/ani9060300_

Round 1
Reviewer 1 Report
The manuscript is well-written, and the procedures/method are described clearly and in an exhaustive manner. The main weaknesses of the paper are the reduced sample size, and the low number of sessions of experiment 2, as recognized by the authors themselves.
Moreover, it seems to me that the gestures used in Experiment 1 (pointing with the target, indicating body part) are quite different from those used in the second experiment and they appear to be easier than those reported in Table 3. I was wondering whether these differences could explain the lack of results in Experiment 2.
Introduction
The study aims should be explained more clearly, specifying the aim of Experiment 1 and 2 rather than describing the study procedure and providing redundant information (L90-94). Similarly, the last paragraph of the introduction, from L95 to L104 should be shortened or replaced with a clear description of the aim of experiment 2.
L77: please provide species name before the reference [32].
L80-L82: I think that authors should answer these questions and explain with a sentence the findings of references [33, 34].
L88: “As of” should be “From”.
L96: “advantage of gestures over words and gazes”.
L96: “to the modality” should be removed.
Material and Methods
The analyses were performed using the ANOVA function. Did the authors verify the normality of the data before deciding to use parametric statistics? Normality tests used should be specified in the manuscript.
L112: “13-year old” should be “13-year-old”.
L179: Why were the correct responses of the experimental conditions rewarded with the clicker sound only? I was wondering whether this could have impacted the animal motivation and concentration during the experiments. What kind of intermittent schedule of reinforcement was used in the experiments? Were the study orangutans used to that kind of schedule before the beginning of the study? I think it is possible that the introduction of the intermittent schedule (or even a different schedule with longer no-reward intervals) could have affected the performance of the individuals, especially in the more complex in Experiment 2.
Results (3.3)
L211-L214: Instead of reporting the same data explained in Table 2, the authors could provide descriptive statistics in this manuscript section (e.g., mean, standard error).
Discussion (3.4)
L250: “That words appear” should be “The fact that words appear”.
Experiment 2
L258: “Why is it that” should be shortened in “Why did subjects mostly rely”.
Results (4.4)
Descriptive statistics could be included to provide more information about the data in the manuscript.
L375: “his” should be “her”.
Discussion (4.5)
L396: “they were to learn” should be “they had to learn”
L406: “pointing-gesture” should be “Pointing-Gesture”.
General discussion
L414: “the importance of positive interactions between”.
L418: “by a given species or individual”.
L419: “to improving” should be “to improve”.
L432-433: “… should mostly rely on the gestural modality” seems to be too strong, even if the authors point out that their sample size is small. This sentence should be replaced with something like “gestural modality seems to be appropriate/efficient … at least with the study species and individuals”.
L437: “it might be that” should be “it is possible that”.
L473: “they take care of” should be “in their care”.
Author Response
The manuscript is well-written, and the procedures/method are described clearly and in an exhaustive manner. The main weaknesses of the paper are the reduced sample size, and the low number of sessions of experiment 2, as recognized by the authors themselves.
We would like to thank Reviewer 1 for her/his positive assessment of the paper, and for acknowledging its strengths and its weaknesses.
Moreover, it seems to me that the gestures used in Experiment 1 (pointing with the target, indicating body part) are quite different from those used in the second experiment and they appear to be easier than those reported in Table 3. I was wondering whether these differences could explain the lack of results in Experiment 2.
This is absolutely true. We now discuss many of the potential sources of the discrepancies between performance in Experiment 1 vs. Experiment 2 (including the one suggested by Reviewer 1) (lines 486-496). We also discuss the general limitations of Experiment 2 by itself (lines 494-501).
Introduction
The study aims should be explained more clearly, specifying the aim of Experiment 1 and 2 rather than describing the study procedure and providing redundant information (L90-94). Similarly, the last paragraph of the introduction, from L95 to L104 should be shortened or replaced with a clear description of the aim of experiment 2.
This is an excellent recommendation. We amended the part of the Introduction that Reviewer 1 mentioned to focus on the aims of both Experiment 1 and 2 (lines 91-97).
L77: please provide species name before the reference [32].
Since this reference is only used as one of the many examples of interspecies communication, we were reluctant to indicate which species were under study in the reference [32]. We added some more references in line with the claim being made instead (see new references 33 and 34).
L80-L82: I think that authors should answer these questions and explain with a sentence the findings of references [33, 34].
This has been done (lines 82-83).
L88: “As of” should be “From”.
This has been changed for “currently”(line 89).
L96: “advantage of gestures over words and gazes”.
Since this part of the Introduction has been modified following a previous comment by Reviewer 1, this part of the sentence does not exist anymore in the revised version of the manuscript.
L96: “to the modality” should be removed.
See our response to the previous comment.
Material and Methods
The analyses were performed using the ANOVA function. Did the authors verify the normality of the data before deciding to use parametric statistics? Normality tests used should be specified in the manuscript.
We would like to thank Reviewer 2 for the careful reading of our manuscript. The Anova function in car package calculates analysis-of-variance tables based on Type-II Wald tests according to the principle of marginality, testing each term after all others. The normality of the data does not need to be verified for the Anova function in car package to be used, since it provides Wald chi-square tests for fixed effects even in generalized linear mixed-effects models, such as those that we have fitted in this study for analysing binomial data.
We made this clearer in the methods section (lines 139-140).
L112: “13-year old” should be “13-year-old”.
This has been corrected (line 104).
L179: Why were the correct responses of the experimental conditions rewarded with the clicker sound only? I was wondering whether this could have impacted the animal motivation and concentration during the experiments. What kind of intermittent schedule of reinforcement was used in the experiments? Were the study orangutans used to that kind of schedule before the beginning of the study? I think it is possible that the introduction of the intermittent schedule (or even a different schedule with longer no-reward intervals) could have affected the performance of the individuals, especially in the more complex in Experiment 2.
We would like to thank Reviewer 1 for this very interesting comment.
The fact that a correct response was rewarded with the clicker sound only indeed differs from classic procedures where many of the good responses are rewarded. The main reason why food was not provided then is that perfect performance (90 successful trials) would have required us to feed the animal 90 times, an amount of food which is beyond the dietary requirement of the animal (this is now explained lines 173-175). To avoid reinforcing one condition over others (if food was provided), we preferred to avoid food rewards altogether. This being said, and since none of the conditions of Experiment 1 gave rise to a food reward after a correct response, we can safely reckon that this schedule does not explain the pattern observed in Experiment 1 (i.e., clear differences between conditions in terms of correct responses). Regarding Experiment 2, it should be reminded that a correct response led to feeding, for as many as 20 pieces of apple per session (lines 332-333).
Results (3.3)
L211-L214: Instead of reporting the same data explained in Table 2, the authors could provide descriptive statistics in this manuscript section (e.g., mean, standard error).
We have added the standard errors (lines 208-212) and removed Table 2 that was redundant indeed.
Discussion (3.4)
L250: “That words appear” should be “The fact that words appear”.
This has been corrected (line 247).
Experiment 2
L258: “Why is it that” should be shortened in “Why did subjects mostly rely”.
This has been changed (line 264).
Results (4.4)
Descriptive statistics could be included to provide more information about the data in the manuscript.
We have added the standard errors (lines 362-365 and 389-390) as we did in Results 3.3.
L375: “his” should be “her”.
This has been changed (line 392).
Discussion (4.5)
L396: “they were to learn” should be “they had to learn”
This has been changed (line 412).
L406: “pointing-gesture” should be “Pointing-Gesture”.
This has been changed for every instance of the expression (lines 422 and 428).
General discussion
L414: “the importance of positive interactions between”.
We corrected this (line 430).
L418: “by a given species or individual”.
We now acknowledge that individual differences should also be taken into account regarding preference for a modality above another (line 435).
L419: “to improving” should be “to improve”.
We corrected this (line 435).
L432-433: “… should mostly rely on the gestural modality” seems to be too strong, even if the authors point out that their sample size is small. This sentence should be replaced with something like “gestural modality seems to be appropriate/efficient … at least with the study species and individuals”.
This is a very fair point that we now acknowledge (lines 449-451).
L437: “it might be that” should be “it is possible that”.
This has been corrected (line 460).
L473: “they take care of” should be “in their care”.
We corrected this (line 511). We are grateful to Reviewer 1 for pointing out this grammatical mistake and others.

Reviewer 2 Report
This is a nice experimental paper that shows for a sample of 2 captive orangutans that gestural signals are more effective than either verbal or gaze directed signals in keepers communicating with the apes. This has value in increasing the quality of interactions between captive animals and their care givers. The main limitations are the small sample size with the two participants actually being related and the fact that the failure to obtain significant results in Experiment 2 that can be explained by procedural issues rather than an inability to learn. Given that orangutans are viewed by many researchers and observers in zoos as being highly intelligent and skilled at finding ways to escape and/or modify their environments, the failure to learn to associate arbitrary words or gestures with novel visual stimuli or to differentiate (or perform well) with iconic versus non-iconic gestures could be explained by the animals not being familiar with the novelty of the task, the limited amount of training, over-shadowing by prior training that involved body parts and iconic gestures, the lack of a time out for incorrect responses, or by lack of consistent food reward for correct responding. As a result while I think Experiment 1 is valid and useful, I do not think Experiment 2 can be interpreted without additional research.
The information given in Table 2 is well described in the text and so table 2 could be deleted and suggest that the paper focus on experiment 1 alone, eliminating experiment 2, especially given that 2 B has only a single orangutan participating. I think it is worthwhile pursuing the goals of Experiment 2 with additional participants and with a methodology that eliminates some of the confounds in the current experiment. For example Schumacher has shown that orangutans can readily learn a large number of arbitrary symbols that are associated with words, and van Schaik and colleagues have reported on examples of cognitive complexity, tool, use and even culture in wild orangutans. So the negative results reported here do not mesh with results in other research.
Finally, some theorists of language origins have posited that language arose from gestural communication so a connection between the results in Experiment 1 and these theories would add to the interest of the paper.
Minor points:
l. 51 “secured” has no meaning in this context. Eliminate.
l. 114 It is more traditional to use X rather than # in describing dimensions of enclosures.
l. 137 should be section 2.4 not 3.4
l. 142 should be section 2.5 not 3.5
l. 240 “but existed prior to our”
l. 285 “positioned” not “scratched”
l. 317 “titled” not “leant” also l. 337
l. 426 “whereas” rather than “while”
Author Response
This is a nice experimental paper that shows for a sample of 2 captive orangutans that gestural signals are more effective than either verbal or gaze directed signals in keepers communicating with the apes. This has value in increasing the quality of interactions between captive animals and their care givers. The main limitations are the small sample size with the two participants actually being related and the fact that the failure to obtain significant results in Experiment 2 that can be explained by procedural issues rather than an inability to learn.
We would like to thank Reviewer 2 for her/his positive assessment of our contribution. Reviewer 2 is right in pointing out that the absence of significant results in Experiment 2 may be due to the experimental procedure, rather than to the inability of orangutans to produce associative learning. We have now made this clearer between lines 486 and 504.
Given that orangutans are viewed by many researchers and observers in zoos as being highly intelligent and skilled at finding ways to escape and/or modify their environments, the failure to learn to associate arbitrary words or gestures with novel visual stimuli or to differentiate (or perform well) with iconic versus non-iconic gestures could be explained by the animals not being familiar with the novelty of the task, the limited amount of training, over-shadowing by prior training that involved body parts and iconic gestures, the lack of a time out for incorrect responses, or by lack of consistent food reward for correct responding. As a result while I think Experiment 1 is valid and useful, I do not think Experiment 2 can be interpreted without additional research.
We now acknowledge that (i) the results of Experiment 2 are at odd with the known capabilities of orangutans (lines 488-489) and that (ii) the interpretation of those results should be made in line with all the limitations pertaining to the procedure, i.e.: the novelty of the task, the fact that there was little opportunity to learn (and the fact that the inconsistent food rewarding made have made this worse), the influence of the previous experimental procedure and the limited amount of time to produce a response. This is all discussed between lines 489 and 496.
The information given in Table 2 is well described in the text and so table 2 could be deleted and suggest that the paper focus on experiment 1 alone, eliminating experiment 2, especially given that 2 B has only a single orangutan participating. I think it is worthwhile pursuing the goals of Experiment 2 with additional participants and with a methodology that eliminates some of the confounds in the current experiment. For example Schumacher has shown that orangutans can readily learn a large number of arbitrary symbols that are associated with words, and van Schaik and colleagues have reported on examples of cognitive complexity, tool, use and even culture in wild orangutans. So the negative results reported here do not mesh with results in other research.
We agreed that Table 2 was superfluous so we deleted it from the main text, and reported the standard errors together with the means in the text directly (lines 208-212).
We believe that we now acknowledge the fact that orangutans are perfectly capable of learning the associations provided in Experiment 2 (see our response to the previous comments). We would like to thank Reviewer 2 for providing two key references here. The second one is now cited at line 488 together with 4 other references (we could not find the Schumacher’s reference and thus we apologize for not having added it to the revised manuscript). Besides, we make recommendations for future usage of procedures akin to the one used in Experiment 2 (lines 501-504).
The suggestion of Reviewer 2 to eliminate Experiment 2 altogether is legitimate, but we believe that our tentative is worth reporting. It will give readers a full picture on what we’ve done (and not solely what has yielded positive results) and help researchers avoid certain of our mistakes when investigating comparable issues in future endeavours. We reckon that Experiment 2 is very tentative: this is again readdressed (line 505), before closing the Discussion section. We hope Reviewer 2 will find our decision to keep Experiment 2 and the discussion of its limitations appropriate.
Finally, some theorists of language origins have posited that language arose from gestural communication so a connection between the results in Experiment 1 and these theories would add to the interest of the paper.
This is a very interesting perspective we now discuss (lines 451-454), in line with failure to teach speech to apes.
Minor points:
l. 51 “secured” has no meaning in this context. Eliminate.
This has been changed for “safe” (line 51).
l. 114 It is more traditional to use X rather than # in describing dimensions of enclosures.
This has been corrected (line 106).
l. 137 should be section 2.4 not 3.4
This has been corrected (line 129).
l. 142 should be section 2.5 not 3.5
This has been corrected (line 134).
l. 240 “but existed prior to our”
This has been corrected (line 237).
l. 285 “positioned” not “scratched”
This has been corrected (line 291).
l. 317 “titled” not “leant” also l. 337
This has been corrected in both occurrences (we guessed Reviewer 2 meant ‘tilted’ – lines 323 and 345).
l. 426 “whereas” rather than “while”
This has been corrected (line 444).
Reviewer 3 Report
This study on two captive apes, assesses the effectiveness of caretaker-animal communication. By assessing how information transfer between caretaker and captive animals can be maximised, the authors argue we will be able to enhance animal welfare. I think the authors have cleverly taken advantage of a real-world training procedure, and have produced some quite convincing data regarding the most successful modality of communication.
In terms of the writing, and layout of the manuscript – the paper reads extremely well. I have a couple of general concerns below, which when addressed, I feel would make the manuscript stronger:
- What does success and failure mean?
It is not clear from the manuscript what exactly constitutes a failed trial. If an orangutan was unsuccessful in a trial, was this because they responded incorrectly, or because they did not respond at all? As these are distinctly different. One is evidence that the orangutan is engaging in a communicative interaction (however, incorrectly interpreting the signal), and one is evidence that no information is being communicated at all. I understand the study has been designed as a binomial (success 1 vs failure 0) – however going into some more detail, at the very least in terms of descriptive statistics about incongruent responses vs. congruent responses may be interesting and informative.
- Differences in the vocal and gestural signals.
Do the authors believe that comparisons between the vocal and gestural signals here are truly fair? Here, the vocal signal changes each time (a different word for each body part), however, the gestural signal remains exactly the same, with only the location of the gesture moving. Can we really agree that these gestures are different to the same extent as the vocal signals are, and if not, can we really make a gestural vs vocal signal argument comfortably?
Intuitively, I believe that if the training procedure required the apes to respond to a completely unique gesture, produced in a single spatial location, (e.g. Thumbs up = present arm, Thumbs down = present leg), the subjects would perform similarly bad as they do with vocal cues alone.
I think this needs to be discussed in the manuscript, and I am interested in the authors thoughts. However, I do not think it completely detracts away from the conclusions – I am aware the authors did not design the training system themselves, and the manuscript still reports an efficient method to communicate to apes in captive settings, it simply weakens the Gestural vs Vocal modality argument.
- Statistics and sample size.
Although I do not claim to be a statistician, I believe there may be a small issue with how the subjects’ identity has been incorporated into the GLMM. In your supplementary scripts file, there is justification as to why subject ID was kept as a fixed effect, rather than incorporating this variable as a random factor (of which I thought did make some sense), however I think doing so may be a little problematic. Many may argue that this data has been pseudo-replicated, and each data point in the model will be treated as independent (in respect to ID) – potentially inflating the results of your other fixed effects.
I can imagine a modelling approach with n=2 may be somewhat tricky, and I am not experienced in working around this. I think however, your decisions to use subjects as a fixed-effect should be more clearly stated in the manuscript and not hidden within the supplementary files as this could cause some concern to readers. Alternatively (and what would most satisfy me personally), if its absolutely impossible to fit your models with subjects as a random factor due to convergence issues, I would consider running the data of each subject separately, and conceding that assessing individual differences may not be possible with this data. Splitting subjects data up into separate analyses is common in studies of animal cognition where sample sizes are very low. The raw data is very convincing, and I do not believe that amending the statistics would change any of findings.
On a related note – it would be beneficial to add a word or two of caution around the sample size issue of the study in the discussion. I think these small sample studies are absolutely helpful to the field, and do not hesitate for recommending such a design from publication, but, it is important the author is clear about the problems and risk of such an approach.
Author Response
This study on two captive apes, assesses the effectiveness of caretaker-animal communication. By assessing how information transfer between caretaker and captive animals can be maximised, the authors argue we will be able to enhance animal welfare. I think the authors have cleverly taken advantage of a real-world training procedure, and have produced some quite convincing data regarding the most successful modality of communication. In terms of the writing, and layout of the manuscript – the paper reads extremely well. I have a couple of general concerns below, which when addressed, I feel would make the manuscript stronger:
We would like to thank Reviewer 3 for her/his positive assessment of our manuscript. We are also grateful to her/him for calling for clarifications, which, we also believe, have made our manuscript much stronger.
- What does success and failure mean?
It is not clear from the manuscript what exactly constitutes a failed trial. If an orangutan was unsuccessful in a trial, was this because they responded incorrectly, or because they did not respond at all? As these are distinctly different. One is evidence that the orangutan is engaging in a communicative interaction (however, incorrectly interpreting the signal), and one is evidence that no information is being communicated at all. I understand the study has been designed as a binomial (success 1 vs failure 0) – however going into some more detail, at the very least in terms of descriptive statistics about incongruent responses vs. congruent responses may be interesting and informative.
We have added details to explain what we considered as successful or unsuccessful trials (lines 329-331, 357-360) and we have provided the percentages of trials in which orangutans subjects did not respond at all as descriptive statistics in the Results section of Experiment 2 (lines 362-365 and 389-390).
- Differences in the vocal and gestural signals.
Do the authors believe that comparisons between the vocal and gestural signals here are truly fair? Here, the vocal signal changes each time (a different word for each body part), however, the gestural signal remains exactly the same, with only the location of the gesture moving. Can we really agree that these gestures are different to the same extent as the vocal signals are, and if not, can we really make a gestural vs vocal signal argument comfortably?
Intuitively, I believe that if the training procedure required the apes to respond to a completely unique gesture, produced in a single spatial location, (e.g. Thumbs up = present arm, Thumbs down = present leg), the subjects would perform similarly bad as they do with vocal cues alone.
I think this needs to be discussed in the manuscript, and I am interested in the authors thoughts. However, I do not think it completely detracts away from the conclusions – I am aware the authors did not design the training system themselves, and the manuscript still reports an efficient method to communicate to apes in captive settings, it simply weakens the Gestural vs Vocal modality argument.
This is a very fair comment. In fact, the gestural modality mostly employs ‘pointing’ which is specifically why we ran phase B of Experiment 2, so as to investigate whether this could explain the pattern observed in Experiment 1 (as explained between lines 276 and 279). In response to this comment, we acknowledge this difference between Gestures and other conditions (lines 253-261). We also indicate that, when selecting only requests for which the signal does not involve pointing (the teeth and the tongue, which involves spreading the thumb and index and sticking the tongue as signals), the same pattern is observed (greater performance for the gestural condition), suggesting that the gestural modality (beyond pointing) may be better understood overall (lines 255-258).
- Statistics and sample size.
Although I do not claim to be a statistician, I believe there may be a small issue with how the subjects’ identity has been incorporated into the GLMM. In your supplementary scripts file, there is justification as to why subject ID was kept as a fixed effect, rather than incorporating this variable as a random factor (of which I thought did make some sense), however I think doing so may be a little problematic. Many may argue that this data has been pseudo-replicated, and each data point in the model will be treated as independent (in respect to ID) – potentially inflating the results of your other fixed effects
I can imagine a modelling approach with n=2 may be somewhat tricky, and I am not experienced in working around this. I think however, your decisions to use subjects as a fixed-effect should be more clearly stated in the manuscript and not hidden within the supplementary files as this could cause some concern to readers. Alternatively (and what would most satisfy me personally), if its absolutely impossible to fit your models with subjects as a random factor due to convergence issues, I would consider running the data of each subject separately, and conceding that assessing individual differences may not be possible with this data. Splitting subjects data up into separate analyses is common in studies of animal cognition where sample sizes are very low. The raw data is very convincing, and I do not believe that amending the statistics would change any of findings.
We agree that dealing with n=2 is quite difficult, especially because it is not correct to fit models with a random factor having only two levels. Details and justifications were provided as supplementary materials but we now indicate in the manuscript too that “we decided to use a modelling approach with Subject as fixed effect and to aggregate the data across sessions and sequences to avoid any risk of pseudo-replication” (lines 140-142). In addition, we state in the general discussion that “Although we used a modelling approach well-suited to analyze responses of 2 individual subjects, statistical analyses would be more powerful with additional subjects” (lines 455-457).
On a related note – it would be beneficial to add a word or two of caution around the sample size issue of the study in the discussion. I think these small sample studies are absolutely helpful to the field, and do not hesitate for recommending such a design from publication, but, it is important the author is clear about the problems and risk of such an approach.
We indicate in the Discussion section that ‘One should note, however, that our study was based on 2 individuals trained at a specific site and with one specific keeper. Ideally, our study should be replicated in other settings and in other sites, possibly with a larger cohort.” (lines 457-458). This is now reminded in the Conclusion (lines 515-516).
Round 2
Reviewer 2 Report
The authors have addressed most prior concerns and I reluctantly accept keeping Exp. 2 in the current manuscript. Although I think it is premature to publish such results, the authors have suitably noted the problems with the study and do call for further research in specific ways.
There are several minor edits that need to be addressed:
l. 115 “orangutan subjects”
l. 119 Theodora is 31 years old in 2019 but was only 29 at the start of the study in 2017. Tamu was 13 at the start of the study but would be 15 this year. The ages should be consistent with a specific year, ideally, the start of the research.
l. 132 “have been involved”
ll. 315-337 Original Table 2 was removed so the references here should all be to Table 2 not table 3 and the title of the table at l. 338 should be Table 2.
Ref 13 has too much detail about the publisher location and material does not need to be in all capital letters. The reference should simply cite the American Journal of Primatology, and the page number.
Ref. 30 use lower case on “meanings”
Ref. 33 should not have key words in the title capitalized
Ref. 31 just lists the author and title but no source. Is this from a journal, a web site, a user’s manual? How can one locate this article?
Ref. 59 Again main words in title are capitalized and should be in lower case
Author Response
Reply to Reviewer 2 – second review :
The authors have addressed most prior concerns and I reluctantly accept keeping Exp. 2 in the current manuscript. Although I think it is premature to publish such results, the authors have suitably noted the problems with the study and do call for further research in specific ways.
We would like to thank Reviewer 2 for her/his positive assessment of our manuscript. We are also grateful to her/him for calling for clarifications, which, we also believe, have made our manuscript much stronger despite limitations of our Exp. 2.
There are several minor edits that need to be addressed:
l. 115 “orangutan subjects”
This has been corrected.
l. 119 Theodora is 31 years old in 2019 but was only 29 at the start of the study in 2017. Tamu was 13 at the start of the study but would be 15 this year. The ages should be consistent with a specific year, ideally, the start of the research.
We have corrected the age of Theodora and explained that ages are “i.e. ages when the study started”.
l. 132 “have been involved”
This has been corrected.
ll. 315-337 Original Table 2 was removed so the references here should all be to Table 2 not table 3 and the title of the table at l. 338 should be Table 2.
Thank you very much for pointing out this mistake. Previous table 3 is now table 2 throughout the manuscript.
Ref 13 has too much detail about the publisher location and material does not need to be in all capital letters. The reference should simply cite the American Journal of Primatology, and the page number.
We took advantage of this comment to revise our citation and cite a more recent work on positive reinforcement in the same species(i.e., Bloomsmith, M.; Neu, K., Franklin, A., Griffis, C., McMillan J. Positive reinforcement methods to train chimpanzees to cooperate with urine collection. J. Am. Assoc. Lab. Anim. Sci. 2015, 54, 66-69).
Ref. 30 use lower case on “meanings”
This has been corrected.
Ref. 33 should not have key words in the title capitalized
This has been corrected.
Ref. 31 just lists the author and title but no source. Is this from a journal, a web site, a user’s manual? How can one locate this article?
We assumed that Reviewer 2 meant ref. 41. This reference points towards a webpage. We corrected the citation to comply with the journal guidelines. This now reads: Emmeans: Estimated marginal means, aka least-squares means. Available online: https://rdrr.io/cran/emmeans/ (accessed on 21 May 2019).
Ref. 59 Again main words in title are capitalized and should be in lower case
This has been corrected.